# A nonrandomized open-label phase 2 trial of nonischemic heart preservation for human heart transplantation

Johan Nilsson [1✉], Victoria Jernryd[1], Guangqi Qin [1], Audrius Paskevicius[1], Carsten Metzsch[1], Trygve Sjöberg[1] & Stig Steen[1]

Pre-clinical heart transplantation studies have shown that ex vivo non-ischemic heart pre-servation (NIHP) can be safely used for 24 h. Here we perform a prospective, open-label, non-randomized phase II study comparing NIHP to static cold preservation (SCS), the current standard for adult heart transplantation. All adult recipients on waiting lists for heart trans-plantation were included in the study, unless they met any exclusion criteria. The same standard acceptance criteria for donor hearts were used in both study arms. NIHP was scheduled in advance based on availability of device and trained team members. The primary endpoint was a composite of survival free of severe primary graft dysfunction, free of ECMO use within 7 days, and free of acute cellular rejection ≥2R within 180 days. Secondary endpoints were I/R-tissue injury, immediate graft function, and adverse events. Of the 31 eligible patients, six were assigned to NIHP and 25 to SCS. The median preservation time was 223 min (IQR, 202–263) for NIHP and 194 min (IQR, 164–223) for SCS. Over the first six months, all of the patients assigned to NIHP achieved event-free survival, compared with 18 of those assigned to SCS (Kaplan-Meier estimate of event free survival 72.0% [95% CI 50.0–86.0%]). CK-MB assessed 6 ± 2 h after ending perfusion was 76 (IQR, 50–101) ng/mL for NIHP compared with 138 (IQR, 72–198) ng/mL for SCS. Four deaths within six months after transplantation and three cardiac-related adverse events were reported in the SCS group compared with no deaths or cardiac-related adverse events in the NIHP group. This first-in-human study shows the feasibility and safety of NIHP for clinical use in heart transplantation. ClinicalTrial.gov, number NCT03150147

[1] Department of Clinical Sciences Lund, Cardiothoracic Surgery, Lund University and Skane University Hospital, Lund, Sweden. ✉email: johan.nilsson@med.lu.se

Survival after heart transplantation (HT) has improved markedly over the past three decades, but graft dysfunction still remains the leading cause of early mortality. Only one-third of all donated hearts are used because of the risk of early and late graft dysfunction or logistical problems due to the limitations of acceptable allograft ischemic time[1,2]. Donors are generally older and have more comorbidities now than before[3]. Allograft ischemia lasting more than 4 h increases the risk of mortality, and marginal donors are less tolerant to ischemia[4,5].

Despite a general improvement in most aspects of HT, donor hearts are still preserved prior to transplantation with an ischemic static cold storage (SCS). Ischemia and reperfusion (I/R) damage contributes to early dysfunction of the donor heart and death of the recipient. Ischemia results in tissue hypoxia and micro-vascular dysfunction[6–8]. The subsequent reperfusion increases the activation of innate and adaptive immune responses, resulting in a cell death program[7,9]. The injured endothelium increases the risk of acute cellular rejection (ACR) and cardiac allograft vas-culopathy (CAV)[10]. Together, these factors affect early and late survival[11,12].

With the SCS method, the heart is flushed with cold crystalloid solutions and transported on ice. The nonischemic heart-preservation (NIHP) system is instead a portable device approved for ground and airborne transportation (Fig. 1)[13]. The heart is continuously perfused with a cold (8 °C) oxygenated cardioplegic nutrition–hormone solution containing erythrocytes from the blood bank. This is in contrast to the organ care system which uses a warm, noncardioplegic preservation solution con-taining donor blood[14].

Preclinical studies, using the NIHP system, have shown that the pig donor heart can be safely preserved for 24 h and that the endothelium contractile function can be preserved for at least 8 h[8,13,15]. In a recently published study of life-supporting porcine cardiac xenotransplantation using the same system, NIHP was one of two keys to the success[16]. Therefore, an NIHP system might allow the procurement of distant donor hearts and possibly enable resuscitation of marginal donor hearts, thereby expanding the donor pool. However, this state-of-the-art technology has never been applied to humans.

Here we report the first-in-human use of the NIHP method in adult HT. In this nonrandomized phase II study, we investigate event-free survival and immediate graft function. We show a decrease of cardiac injury markers, less ACR, and no death or cardiac-related serious adverse events among recipients trans-planted using the NIHP method. Our results show that NIHP is safe and feasible, encouraging further clinical investigations.

## Results

**Recruitment.** Between April 2, 2017 and September 25, 2018, 42 patients underwent HT, 11 patients were excluded because they met one of the exclusions criteria (4 patients), did not provide written informed consent (4 patients), or required an urgent transplantation (3 patients). Transplantation was planned in advance when the NIHP method could be used because the device and team members trained to use the system must be available. This resulted in the NIHP system being assigned to 6 patients out of the total 31 eligible patients (Fig. 2). The donor and recipient's characteristics did not exclude any patient from being assigned to the NIHP group; however, they were excluded if they met any exclusion criteria. Following organ retrieval, all organs were used. All patients were followed-up for 6 months or until death, and no data on outcomes were missing. The latest follow-up occurred on March 25, 2019.

**Donor, recipient, and preservation characteristics.** Table 1 shows the baseline characteristics of the donors and recipients in the two study groups. Overall, eight (26%) recipients and nine (29%) donors were women. The median age was 54 years (interquartile range [IQR], 43–60) for the donors and 56 years (IQR, 46–64) for the recipients. Baseline characteristics, except for body size, were similar for those in the two groups. The donor size was similar in the two groups but the NIHP recipients were larger and had a median body mass index (BMI) of 30 kg/m² (IQR, 29–32) compared with the SCS group, who had a median BMI of 26 kg/m² (IQR, 23–28). This resulted in a larger and

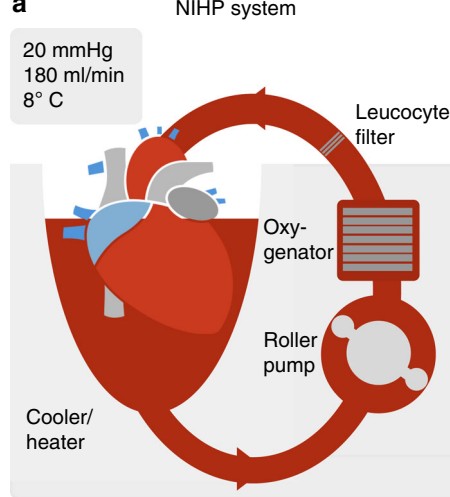

**a** NIHP system

20 mmHg
180 ml/min
8° C

Leucocyte filter

Oxy-genator

Roller pump

Cooler/heater

**b** Heart preservation solution

| | |
|---|---|
| Sodium (Na⁺) | 136 mmol/L |
| Potassium (K⁺) | 23 mmol/L |
| Calcium (Ca²⁺) | 1.3 mmol/L |
| Magnesium (Mg²⁺) | 8.0 mmol/L |
| Chloride (Cl⁻) | 142 mmol/L |
| Bicarbonate (HCO₃⁻) | 25 mmol/L |
| Phosphate (PO₄²⁻) | 1.3 mmol/L |
| D-Glucose | 6.3 mmol/L |
| Albumin | 75 g/L |
| Dextran-40 | 1 g/L |
| Cocaine | 6 nmol/L |
| Noradrenaline | 6 nmol/L |
| Adrenaline | 6 nmol/L |
| Triiodothyronine (T3) | 3 nmol/L |
| Cortisol | 420 nmol/L |
| Insulin | 8 U/L |
| Imipenem | 20 mg/L |
| Erythrocytes (Hct) | 15% |
| 95% O₂ + 5% CO₂ | 0.2 L/min |

**c** Heart mounting

**Fig. 1 The nonischemic heart-preservation method (NIHP).** Shown is a drawing of the NIHP method (**a**). The equipment consists of a reservoir, a pressure-controlled roller pump, an oxygenator, an arterial-leukocyte filter, a heater–cooler unit, oxygen and carbon dioxide containers, a gas mixer, sensors, and a programmable control system. The reservoir is filled with 2.5 L of the perfusion solution (**b**) plus ~500 mL compatible irradiated and leukocyte-reduced blood cells from the hospital blood bank, providing a hematocrit of ~15%. Perfusion is provided through the aortic cannula to the coronary vessels. The picture (**c**) shows the first human heart transplantation using the NIHP method. The heart is mounted and submerged into the heart-preservation solution, which is actively regulated to maintain a pH of ~7.4 and a temperature of 8 °C. The device software is adjusted to maintain a mean blood pressure of 20 mmHg in the aortic root, providing a coronary flow between 150 and 250 mL/min.

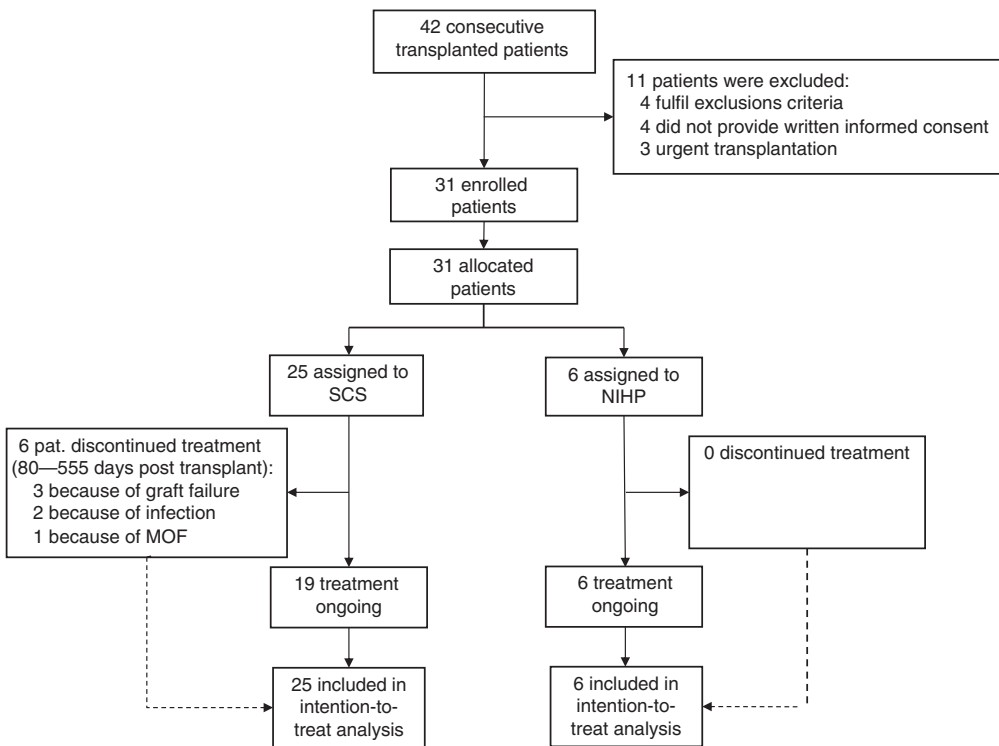

**Fig. 2 CONSORT flow diagram.** Modified CONSORT flow diagram for all recipients enrolled in the trial. MOF multi-organ failure; NIHP nonischemic heart preservation; SCS static cold storage.

unfavorable size mismatch in the NIHP group (recipient/donor BMI, 1.2; IQR, 1.1–1.4) compared with the SCS group (recipient/donor BMI, 0.9; IQR 0.7–1.1). Furthermore, the median total preservation time was longer for the NIHP group (223 min; IQR, 202–263) than for the SCS group (194 min; IQR, 164–223), (Supplementary Fig. 1).

**Ex-vivo perfusion data.** We arrested the donor hearts in the NIHP group with the heart-preservation solution without erythrocytes. Then, we harvested the hearts in the same way as performed for the SCS group. We cannulated the distal ascending aorta from the device and submerged the heart in the preservation medium (Fig. 1c). The median preperfusion organ mounting time (ischemic time) was 24 min (IQR, 20–28 min) (Supplementary Fig. 2). The organ was perfused for a median 140 min (IQR, 109–162 min) with a pressure of 20 mmHg (IQR, 19–21 mmHg) resulting in coronary blood flow of 178 mL/min (IQR, 160–221 mL/min). The temperature was stable at 8 °C during the entire perfusion time (Supplementary Fig. 3). The median aB-lactate was 1.5 mmol/L preperfusion (IQR, 1.2–1.5) and 1.4 mmol/L (IQR, 1.3–1.5) after continuous perfusion (Supplementary Table 1).

**Event-free graft survival (primary outcome).** During the first 6 months, all of the patients assigned to the NIHP group met the primary composite outcome of event-free survival (survival free of severe primary graft dysfunction (PGD) at 24 h, free of extracorporal mechanical support use within 7 days, and free of ACR ≥ 2R within 180 days); however, only 18 (72%) of those assigned to the SCS group achieved event-free survival (Kaplan–Meier estimate of event-free survival 72%; 95% confidence interval (CI), 50–86%) (Table 2 and Fig. 3). All patients survived the first 30 days after transplantation. No death or cardiac-related serious adverse events were reported within 6 months after transplantation in the NIHP group; however, four

(16%) death and three (12%) cardiac-related serious adverse events occurred in the SCS group (Table 3).

**Secondary outcomes of NIHP and SCS group.** Although the NIHP group had a longer duration of preservation (out of body) and the recipients were matched with smaller donors compared with the SCS group, we did not observe any difference in terms of early organ dysfunction or the need for inotropic support. As shown in Table 2, the immediate graft function was similar for both groups. However, there was a difference in cardiac injury markers. One patient (20%) in the NIHP group had a pathological cardiac troponin I (cTnI) > 0.02 ng/mL at the end of preservation compared with all patients in the SCS group (Table 2). Furthermore, the median creatine kinase-muscle/brain (CK-MB) level, assessed 6 ± 2 h after ending perfusion were 76 ng/mL (IQR, 50–101) ng/mL for the NIHP group and 138 ng/mL (IQR, 72–198) for the SCS group (Fig. 4).

All patients followed a predefined protocol for surveillance and monitoring. During the first 6 months after transplantation, 2 patients (33%) in the NIHP group had an ACR ≥ 1R; however, 15 patients (60%) in the SCS group did (Supplementary Fig. 4). None of the patients in the NIHP group had an ACR ≥ 2R; however, 4 patients (16%) in the SCS group did.

The NIHP group showed a tendency for reduced postoperative renal function compared with the SCS group (Table 2). The minimum creatinine clearance levels within 7 days after transplantation were 33 mmol/L (IQR, 31–40) for the NIHP group and 44 mmol/L (IQR, 34–59) for the SCS group. Half of the patients in the NIHP group needed continuous renal replacement therapy (CRRT) within the first 7 days after transplantation; however, only six (24%) patients in the SCS group needed CRRT. None of the patients required dialysis treatment at the last follow-up date. The median aspartate aminotransferase (ASAT) on postoperative day 1 was 1.6 (IQR 1.4–2.0) for the NIHP group;

| Table 1 Donor, recipient, and transplantation characteristics. | | |
|---|---|---|
| Donor characteristics | NIHP (n = 6) | SCS (n = 25) |
| Median age (year) | 56 (46-68) | 53 (41-58) |
| Female sex | 1 (17%) | 8 (32%) |
| Median body mass index (kg/m$^2$) | 24 (22-27) | 26 (23-33) |
| Cause of death | | |
| Cerebrovascular event | 2 (33%) | 16 (64%) |
| Head trauma | 3 (50%) | 1 (4%) |
| Other | 1 (17%) | 8 (32%) |
| Blood group | | |
| A | 4 (66%) | 7 (28%) |
| AB | 0 | 3 (12%) |
| B | 1 (17%) | 4 (16%) |
| O | 1 (17%) | 11 (44%) |
| History of smoking | 2 (33%) | 7 (37%) |
| Hypertension | 2 (33%) | 9 (39%) |
| Cytomegalovirus | 6 (100%) | 19 (79%) |
| Recipient characteristics | NIHP (n = 6) | SCS (n = 25) |
| Median age (year) | 59 (56-64) | 55 (46-63) |
| Female sex | 0 | 8 (32%) |
| Median duration on waiting list (days) | 118 (105-222) | 114 (33-340) |
| Median body mass index (kg/m$^2$) | 30 (29-32) | 26 (23-28) |
| Diagnosis | | |
| Ischemic cardiomyopathy | 2 (33%) | 5 (20%) |
| Nonischemic cardiomyopathy | 4 (67%) | 16 (64%) |
| Other | 0 | 4 (16%) |
| Blood group | | |
| A | 5 (83%) | 7 (28%) |
| AB | 0 | 4 (16%) |
| B | 1 (17%) | 4 (16%) |
| O | 0 | 10 (40%) |
| Insulin-treated diabetes | 1 (17%) | 3 (12%) |
| Peripheral vascular disease | 1 (17%) | 5 (20%) |
| History of stroke | 1 (20%) | 1 (4%) |
| Preoperative cytomegalovirus | 4 (67%) | 17 (68%) |
| Median most recent creatinine (µmol/L) | 111 (104-136) | 95 (84-129) |
| Median S-bilirubin (µmol/L) | 11 (4.0-14) | 12 (7.0-23) |
| Median pulmonary vascular resistance (Wood units) | 2.4 (1.7-2.8) | 2.0 (1.7-2.2) |
| Panel-reactive antibody level | | |
| 0-10% | 3 (50%) | 9 (36%) |
| 11-80% | 2 (33%) | 10 (40%) |
| >80% | 1 (17%) | 6 (24%) |
| Ventricular assist device | 3 (50%) | 12 (48%) |
| Transplantation details | NIHP (n = 6) | SCS (n = 25) |
| Median volume cardioplegia (L) | 1.2 (1.1-1.3) | 1.9 (1.8-2.0) |
| Median total preservation time (minutes) | 223 (202-263) | 194 (164-223) |
| Median R/D body mass index ratio | 1.2 (1.1-1.4) | 0.9 (0.7-1.1) |
| Female donor to male recipient | 1 (17%) | 1 (4%) |

Data are n (%) or median (IQR).
R/D recipient/donor, IQR interquartile range.

however, it was 2.6 (IQR, 2.2–3.6) for the SCS group (Table 2). None of the patients developed severe liver failure.

**Serious adverse events**. The proportion of patients who had serious adverse events (cardiac, renal, pulmonary failure, bleeding complication, or the need for a permanent pacemaker) leading to an extended length of stay in the intensive care unit (ICU) were comparable for the two groups (Table 3). The most common adverse events in the two groups were acute renal failure (22 patients; 71%) and respiratory failure (defined as need for a ventilator for more than 48 h) (10 patients; 32%). The length of stay in the ICU was similar for both groups; 7.0 days (IQR,

5.4–17 days) and 6.0 days (5.1–11 days) for the NIHP and SCS groups, respectively (Table 2).

## Discussion

This first-in-human study shows the feasibility and safety of NIHP method's for HT. All patients in the NIHP group had an event-free survival at 6 months; however, only 72% of the patients in the SCS group had event-free survival at 6 months. Among NIHP patients, we did not observe any early mortality or cardiac-related serious complications. However, in the SCS group, three patients received extracorporeal membrane oxygenation and four patients had a moderate ACR.

The pathogenesis of PGD is still unclear, but ischemia/reperfusion injury has been identified as a contributing risk factor[5,17,18]. In the present study, we found a decrease in the cardiac injury marker cTnI obtained immediately after preservation and in CK-MB levels after 6 h in the NIHP group. Troponin and CK-MB are sensitive markers of cardiac ischemia and myocardial damage[6,19]. Preclinical studies have shown that CK-MB levels correlate with ischemia/reperfusion tissue damage with HT. Schecter et al. reported that an increased level of cTnI in the preservation solution is associated with development of PGD[17,20,21]. These findings might indicate that the NIHP method reduces the myocardial damage better than the SCS method.

During the first 6 months after transplantation, we also observed less ACR in the NIHP group than in the SCS group. Decreased allograft rejection may suggest that the endothelium was less damaged in the NIHP group; this has been demonstrated in preclinical studies, and is attributable to less ischemia/reperfusion injuries[8,15,22]. According to the latest International Society for Heart and Lung Transplantation registry report, treatment for rejection within the first year after transplantation was associated with an increased risk of CAV development and an increased mortality risk of up to 50% at 5 years[12]. Furthermore, ischemia/reperfusion tissue injury may enhance the activation of innate and adaptive immune responses, resulting in the initiation of a cell death program[7,9].

We noted more undersized and older donors in the NIHP group than in the SCS group. Unfavorable body size mismatch and older donors are well-known risk factors for PGD[18]. This observation may indicate that using nonischemic preservation for marginal donor hearts can make it possible to expand the donor pool in the future, which has been suggested by others[23,24]. However, a larger study is needed to confirm this observation.

The NIHP method is a new type of technology for clinical use; therefore, a learning effect should be expected. However, all accepted donors were utilized and there were no device-related complications. Both groups had similar proportions of patients with serious adverse events leading to an extended length of stay in the hospital. The simplicity of the NIHP system is probably a significant contribution to these observations. An additional advantage of the method is its hypothermic environment for the heart. The hypothermic preservation provides increased safety and protection against external impacts on the system such as power failure. With normothermic preservation, an interruption in ex-vivo perfusion can result in irreversible damage to the heart. During the only randomized controlled trial evaluating normothermic preservation, five donor hearts were considered unacceptable for transplantation after the use of that preservation system[14]. Because these hearts were considered acceptable initially, it cannot be ruled out that something happened in transit that rendered these hearts unusable. Creating an artificial environment similar to the physiological state in which a warm beating heart is supposed to work is both complicated and risky. Moreover, it involves additional surgical and technical support and

**Table 2 Patients outcomes.**

| Primary outcome | NIHP (n = 6) | SCS (n = 25) | RR/ES (95% CI) |
|---|---|---|---|
| Survival free of event within 180 days | 6 (100%) | 18 (72%) | 1.4 (1.1–1.8) |
| First event that resulted in failure to reach the primary end point | | | |
| PGD within 24 h | 0 | 2 (8%) | – |
| ECMO within 7 days | 0 | 1 (4%) | – |
| ACR ≥ 2R within 180 days | 0 | 3 (12%) | – |
| Death within 180 days | 0 | 1 (4%) | – |
| Secondary outcomes | NIHP (n = 6) | SCS (n = 25) | RR/ES (95% CI) |
| Immediate graft function | | | |
| Reperfusion time (minutes) | 91 (83–95) | 89 (77–107) | 0.14 (−0.75 to 1.03) |
| Inotropic score at 6 h post transplantation | 21 (9–24) | 30 (20–54) | −0.55 (−1.5 to 0.40) |
| LVEF < 40% within 24 h | 0 | 2 (9%) | — |
| RVEF < 40% within 24 h | 1 (17%) | 6 (27%) | 0.61 (0.090–4.1) |
| I/R-tissue injury | | | |
| cTnI > 0.02 ng/mL at end of preservation | 1 (20%) | 15 (100%) | 0.20 (0.035–1.2) |
| CK-MB > 4.3 ng/mL at end of preservation | 0 | 6 (33%) | – |
| CK-MB 6 ± 2 h after ending preservation (ng/mL) | 76 (54–101) | 138 (72–198) | −1.18 (−2.2 to 0.10) |
| CK-MB 12 ± 4 h after ending preservation (ng/mL) | 38 (30–67) | 53 (41–77) | −0.84 (−1.8 to 0.16) |
| CK-MB 24 ± 6 h after ending preservation (ng/mL) | 16 (10–24) | 15 (12–38) | −0.41 (−1.3 to 0.51) |
| Renal function | | | |
| Minimum creatinine clearance within 7 days | 33 (31–40) | 44 (34–59) | −0.83 (−1.8 to 0.18) |
| CRRT within 7 days | 3 (50%) | 4 (16%) | 3.1 (0.94–10) |
| Liver function | | | |
| ASAT within 48 h (μkat/L) | 1.6 (1.4–2.1) | 2.6 (2.2–3.6) | −1.3 (−2.3 to −0.19) |
| ALAT within 48 h (μkat/L) | 0.4 (0.3–0.5) | 0.6 (0.4–0.8) | −1.0 (−2.0 to −0.067) |
| Time on ventilator (hours) | 32 (22–54) | 39 (22–52) | −0.19 (−1.1 to 0.71) |
| Acute rejection (ACR ≥ 1R) within 180 days | 2 (33%) | 15 (63%) | 0.56 (0.17–1.8) |
| Duration of ICU stays (days) | 7.0 (5.4–17) | 6.0 (5.1–11) | 0.062 (−0.81 to 0.95) |

Data are n (%) or median (IQR).
ACR acute cellular rejection, ALAT alanine transaminase, ASAT aspartate aminotransferase, cTnI cardiac troponin I, CK-MB creatinine kinase-muscle/brain, CRRT continuous renal replacement therapy, ECMO extracorporeal membrane oxygenation, ES effect size, ICU intensive care unit, IQR interquartile range, I/R ischemia and reperfusion, LVEF left ventricular ejection fraction, NIHP nonischemic heart preservation, RR relative risk, RVEF, right ventricular ejection fraction, SCS static cold storage.

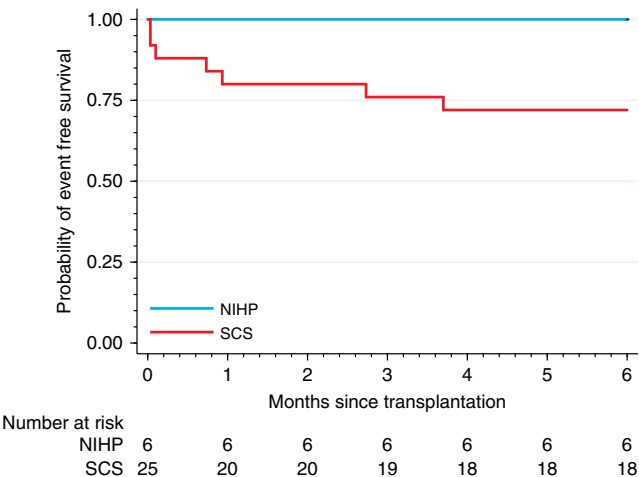

**Fig. 3 The probability of event-free survival.** The Kaplan–Meier plot shows the probability of event-free survival (primary end point) defined as survival free of severe primary graft dysfunction at 24 h, survival free of extracorporeal mechanical support use at 7 days, and survival free of acute cellular rejection ≥2R at 180 days (cyan: NIHP group; red: SCS group). Kaplan–Meier estimate free of event was 72% [95% CI 50–86%] for the SCS group. NIHP (n = 6) nonischemic heart preservation; SCS (n = 25) static cold storage.

appropriate transport, inevitably resulting in more expensive management compared with what is needed for SCS. The future commercial NIHP system will not require extra personnel support.

The potential benefits with NIHP system are an improved postoperative course and reduced total cost of the transplantation. Complications directly connected to the transplant result in increased costs; for example, if the recipient develops PGD requiring mechanical circulatory support, then the ICU stay will be prolonged. An extension of the allograft preservation time will make it possible to schedule transplantation during the day, when the highest competence will be available for these complex, high-risk cases. Furthermore, NIHP may make it possible to increase the donor pool by utilizing more marginal donors and enabling organ sharing across long distances (perhaps even between continents)[25]. Finally, a preservation system that can decrease the activation of innate and adaptive immune responses resulting in a downregulation of the immune system, might provide further benefits for organ transplantation[7,9,24]. This would most likely reduce the need for immunosuppression and decrease the occurrence of complications (for example, toxicity, infection, and malignancies).

Our study has some limitations. Because it was a non-randomized trial, bias in the selection of both donors and recipients could have affected the results. Another limitation of this study was its unblinded nature. Personnel involved in patient care could have favored the innovative NIHP treatment or favored the established SCS technique, thus leaving the direction of the potential bias open to speculation.

In conclusion, this first-in-human study describes the clinical evaluation of a new technology for HT. It represents a first, necessary step in demonstrating that NIHP is feasible, safe, and effective in clinical practice. Because all patients in the NIHP group had an event-free survival at 6 months, further clinical investigations on the efficacy of machine perfusion in HT are warranted[26]. To confirm and extend the results of this study, a randomized trial is required and has been initiated.

## Methods

**Study design.** This investigator-led prospective, open-label, nonrandomized trial of NIHP treatment of donor hearts for HT was performed at Skåne University Hospital, Karolinska University Hospital, Linköping University Hospital, and Uppsala University Hospital, which cover two-thirds of the counties in Sweden. Six patients were permitted to be transplanted with donor hearts preserved with the

**Table 3 Serious adverse events.**

| Serious adverse events | NIHP (n = 6) | SCS (n = 25) | RR |
|---|---|---|---|
| Acute cardiac failure | 0 | 3 (12%) | – |
| Acute bleeding (BARC type IV)[a] | 0 | 6 (24%) | – |
| Respiratory failure | 2 (33%) | 8 (32%) | 1.04 (0.29–3.7) |
| Acute kidney failure (KDIGO)[a] | 5 (83%) | 17 (68%) | 1.23 (0.78–1.9) |
| Acute liver failure | 0 | 0 | – |
| Permanent stroke | 0 | 0 | – |
| Permanent pacemaker | 0 | 4 (16%) | – |

Data are n (%).
NIHP nonischemic heart preservation, RR relative risk, SCS static cold storage.
[a]For definition see online methods.

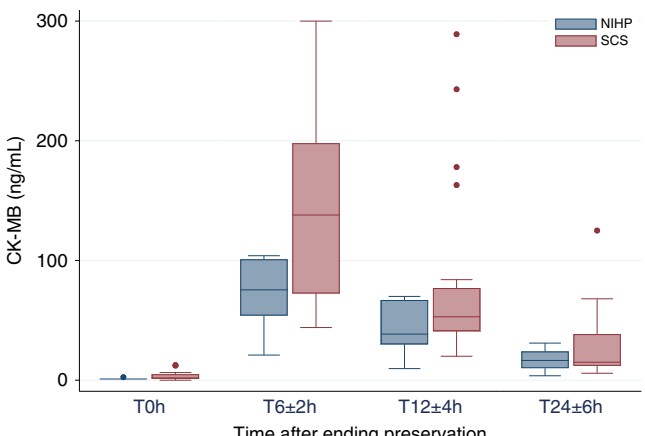

**Fig. 4 Creatine kinase-muscle/brain level after preservation according to the treatment group.** The box plot shows the creatine kinase-muscle/brain (CK-MB) level at different timepoints after ending preservation ($T_0$). Data are represented as boxplots. The middle line is the median, the lower and upper axis correspond to the first and third quartiles, the upper whisker extends from the axis to the largest value no further than 1.5 × interquartile range (IQR) from the axis, and the lower whisker extends from the axis to the smallest value (at most 1.5 × IQR) of the axis. Data beyond the end of the whiskers are outlying points that are plotted individually. NIHP (n = 6) nonischemic heart preservation; SCS (n = 25) static cold storage.

nonischemic method. These were compared with contemporary control patients transplanted with hearts preserved according to standard procedures of SCS. The Swedish research ethics committee approved the trial (2016/603). Patients at the aforementioned centers underwent transplantation at Skane University Hospital; then, they returned to their initial centers for care after transplantation. Transplant candidates were discussed by transplant board members and cardiologists from the participating clinics, and patients accepted for transplantation were screened for the study. Patients accepted for transplantation, who did not fulfill any exclusion criteria, were included in this study after they signed written informed consent. Furthermore, patients on the waiting list were screened for the study (starting April 1, 2017) and those eligible were contacted and included after they signed written informed consent. Due to the delay in the trial registration, a control patient underwent transplantation before the clinical trial registration (ClinicalTrial.gov, number NCT03150147) was completed. The transplantation procedure and perioperative care were completed according to standard practices[27]. All patients were treated with antithymocyte globulin as induction therapy and triple immunosuppression (tacrolimus, mycophenolate mofetil, and glucocorticoids) as maintenance therapy. All participating hospitals followed the national protocol for surveillance and monitoring, which normally includes 14 visits for endomyocardial biopsies during the first year. The biopsies obtained (normally 3–5 biopsies) are sent for histologic evaluation. The grading of ACR (0R, 1R, 2R, or 3R) is done on the basis of an overall assessment of the biopsies according to the ISHLT guidelines[28]. No major amendments were made to the trial design after the start of recruitment.

**Eligibility and consent**. Organ donors had to be 70 years or younger. Donors were excluded if any of the following criteria were fulfilled: insulin-treated diabetes, significant coronary artery disease, hepatitis B-positive or hepatitis C-positive serology; human immunodeficiency virus-positive serology; tuberculosis, malignancy; and abnormal ventricular function < 45%. All adult (aged 18 years or older) recipients on our waiting list for HT were eligible; however, we excluded those who previously underwent solid organ or bone marrow transplantation, had grown up congenital heart disease, had undergone four or more sternotomies, had known malignancy, had kidney failure (Iohexol plasma clearance < 30 at listing), had liver failure (ASAT, alanine transaminase, or total bilirubin more than five-times the upper limit of normal, or international normalized ratio > 2.0), had ongoing septicemia, and had urgent, and/or systemic inflammatory disorders treated with corticosteroids. Potential participants provided consented while on the waiting list, and that consent was affirmed on the day of transplantation. Consent included allowing the recording of anonymized data for trial purposes and the collection of biological samples for storage in the trial biobank.

**Study logistics**. Donor hearts were offered to our heart transplant program through Scandiatransplant (http://www.scandiatransplant.org). Assessment of potential donor hearts were based on the usual constellation of clinical factors, including history, coronary angiography, echo assessment, and direct examination of the heart during procurement. The same standard criteria for donor hearts were used for the NIHP group. Transplantation was scheduled in advance when the NIHP method could be used, because the device and team members trained to use the system must be available. Donors and recipients were excluded from the NIHP method only if they met any of the exclusion criteria. Furthermore, initially, we could only use ground transportation which limited the pool of potential donors who could be assigned to the NIHP method. After the NIHP method was approved for air transport in April 2018, the system could be used without this restriction, which resulting in that the NIHP method being used for 5 of the total 11 transplantations performed over the next 5 months. In addition, as mandated by the local research ethics committees, safety and logistic feasibility were assessed after the first and third patients were subjected to the NIHP method. All patients eligible for transplantation, but not assigned to the NIHP method who had signed the written informed consent and did not fulfill any exclusion criteria, were included as controls during the study period.

**Nonischemic heart-preservation device**. The device used during this study was made in-house and accepted for use by the Department of Medical Technology of Skane University Hospital in Lund, Sweden. The XVIVO Perfusion AB (Göteborg, Sweden) bought the patent to the device and will continue its development with the aim of making it a commercially available device. The device comprises a miniaturized and fully automated heart–lung machine, housed in a portable apparatus (height, 455 mm; length with handles 695 mm; width 415 mm; weight 32 kg), that enables transportation between hospitals (Supplementary Fig. 5). The equipment consists of a reservoir, a pressure-controlled roller pump, an oxygenator, an arterial-leukocyte filter, a heater–cooler unit, oxygen and carbon dioxide containers, a gas mixer, sensors, and a programmable control system. The reservoir (not shown in the figure) is filled with 2.5 L of the heart perfusion solution plus ~500 mL of compatible irradiated and leukocyte-reduced blood cells from the hospital blood bank, providing a hematocrit level of ~15 % (Fig. 1b). The NIHP device software is adjusted to maintain a mean blood pressure of 20 mmHg in the aortic root, providing a coronary flow between 150 and 250 mL/min (Fig. 1a).

**Nonischemic heart-preservation group**. The donor heart was arrested with the heart-preservation solution without erythrocytes (1200 mL) (Fig. 1b). Then the donor heart was then harvested using the same procedure as that used for the SCS group. Thereafter the distal ascending aorta was cannulated from the device with a special double-lumen cannula for easy dearing (Fig. 2a) and a soft 3/8-inch silicon tube was placed into the left ventricle through the atrium to maintain the ventricle in a decompressed state. This was performed to prevent inflation of the left ventricle if leakage of perfusate medium through the aortic valve were to occur during perfusion. The venae cavae and pulmonary artery were left open for a free outlet of perfusate from the coronary sinus. The double-lumen cannula supplying the aorta

with the preservation medium was fixed in a vertical position and the heart was completely submerged in the preservation medium (Fig. 1c). Throughout the perfusion process with the NIHP device, the temperature, perfusion flow, and aortic root pressure were continuously monitored with the built-in sensors. During perfusion of the donor heart (NIHP group) blood samples were retrieved from the reservoir every $30 \pm 10$ min. After explantation of the recipient heart, the continuous perfusion was switched to intermittent perfusion. During the implantation of the heart, the aortic cannula was kept in the aortic root, thereby facilitating stability of the heart. Intermittent perfusions with 200–300 mL of the preservation solution was administrated through the cannula every 15 min during the implantation procedure to avoid ischemia. The cannula was withdrawn before the aortic anastomosis was performed. Blood samples were retrieved from the coronary sinus in the right atrium.

When the NIHP device was used, a research fellow and a research engineer participate in the procedure. The research fellow and research engineer transported the machine-perfusion device to the donor hospital and assisted donor surgeons with connecting the heart to the machine. One of the senior staff surgeons performed the transplantation, and an attending surgeon performed the donor harvesting. No changes were made to the existing rules for organ allocation or transportation protocols.

**Static cold storage group**. For the SCS group, the donor heart was arrested with a crystalloid cardioplegic (1–2 L) solution (Plegisol; Pfizer, New York, NY). The heart was then stored on ice slush at a temperature of ~4 °C. On arrival to the hospital, 500–800 mL of blood cardioplegia was administered to the donor heart, and blood samples from the coronary sinus were obtained and analyzed as described previously.

**Study outcomes**. The primary end point was a composite of survival free of severe PGD at 24 h, free of extracorporal mechanical support use within 7 days, and free of ACR ≥ 2R within 180 days[5,28]. Secondary endpoints included the following: (1) ischemia/reperfusion tissue injury—differences in cTnI and CK-MB collected at end of preservation and $6 \pm 2$, $12 \pm 4$, and $24 \pm 6$ h after the end of preservation (Triage CARDIO3, Alere with Biosite Triage®MeterPro); (2) immediate graft function as indicated by any one of the following clinical indicators: (i) the need for inotropic support (as judged by inotrope score[5]) in the first 6 h after arrival to the ICU, (ii) reperfusion time (time from aortic cross-clamp release in the recipient to termination of cardio pulmonary bypass), (iii) left ventricular ejection fraction (EF) < 40% on days 1 post operatively, (iv) right ventricular EF < 40% on days 1 post operatively; (3) postoperative renal function (difference in estimated minimum creatinine clearance within 7 days post transplant and need for CRRT within 7 days after transplantation); (4) postoperative liver function, peak ASAT and peak alanine transaminase within 24 h after transplantation; (5) postoperative pulmonary function and ventilator requirement (number of hours); (6) ACR ≥ 1R within 6 months after transplantation; (7) length of stay in the ICU; (8) graft and patient survival at 6 months.

During the study, we monitored recipient and donor demographics, medical history, vital signs, laboratory assessments, echocardiography, and right-sided cardiac catheterization. The volumes of the cardioplegic and preservation solutions were registered, as were total preservation and ischemic times. We defined the total preservation time as the donor heart's out-of-body time of the donor heart (i.e., x-clamp on the donor aorta at donor hospital until release of x-clamp donor aorta at transplant center). Cold ischemia time refers to the length of time that the donor heart was kept cold without any continuous perfusion. The main endpoints, PGD and ACR, were blindly assessed.

All endpoints described were included in the current trial registration and were prespecified in the study protocol, except for CK-MB and cTnI collected at end of preservation; these were added to the protocol on September 10, 2017. The timeframe for the primary end point was extended from 30 to 180 days because no events were observed in the NIHP group (study protocol update December 31, 2018). The collection of biological samples from the donor hearts for storage in the trial biobank has not yet been analyzed. Measurements of troponin postoperatively and CK-MB at two extra timepoints, cell-free donor DNA, and EQ-5D were added to the trial registration (NCT03150147) after completion of the nonrandomized part of this study.

**Serious adverse events**. (1) Acute cardiac-related events were defined as the need for an intra-aortic balloon pump and/or mechanical circulatory support within 7 days post transplantation; (2) acute bleeding was defined according to the Bleeding Academic Research Consortium (BARC) type IV criteria (>2000 mL/24 h and/or requiring re-operation for bleeding, and/or intracranial bleeding, and/or transfusion of >5 red blood cell concentrates/48 h)[29]; (3) respiratory failure was defined as impairment of respiratory function requiring re-intubation, requiring tracheostomy, or the inability to discontinue invasive ventilator support within 48 h after cardio pulmonary bypass due to respiratory issues and not due to sedation issues; (4) acute kidney failure was defined according to the Kidney Disease Improving Global Outcomes (KDIGO) criteria as an increase in serum creatinine of >27 μmol/L within 48 h or 1.5 times baseline within 7 days[30]; (5) acute liver failure was defined as the rapid development of hepatocellular dysfunction, specifically coagulopathy, and mental status changes (encephalopathy) in a patient

without prior known liver disease; (6) permanent stroke was defined as an episode of a computed tomography-verified acute neurological dysfunction to be caused by ischemia or hemorrhage that persisted ≥24 h or until death; (7) permanent pacemaker was defined as need for a permanent pacemaker implantation 2 weeks after transplantation.

**Statistical analysis**. The primary outcome (actuarial survival free of event) was analysed using the Kaplan–Meier method. The Kaplan–Meier estimate is presented with 95% CIs. For patients who had more than one event during follow-up that resulted in failure to reach the primary end point, the event that occurred first is the one included in the analysis. The relative risk and 95% CI were calculated for the outcome variables. The effective size was used to compare mean values. Data were assumed to have unequal variances and the approximate degree of freedom was obtained from Welch's formula. Furthermore, continuous variables were log-transformed to fulfill normality assumptions. The baseline value was defined as the last assessment prior to the transplantation. Continuous variables were summarized using the median, and the IQR and categorical variables were summarized using frequency and percentage. Missing values were not imputed. Because of the small sample size in both groups, only descriptive statistics were performed. Data were collected in Microsoft Excel 16.35 (2019 Microsoft Corporation, Redmond, WA). Statistical analyses were performed using Stata MP statistical package version 16.0 (2019 StataCorp LP, College Station, TX).

**Reporting summary**. Further information on research design is available in the Nature Research Reporting Summary linked to this article.

## Data availability

Raw data cannot be shared publicly because of legal and ethical restrictions associated with patient confidentiality. Raw data are available to all interested researchers upon request addressed to the corresponding author J.N. Instructions on how to apply and criteria for access to confidential data are available on the Swedish Ethical Review Authority website (http://etikprovning.se).

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

## Acknowledgements

The authors gratefully acknowledge the important contribution made to this paper by the late professor Peter Höglund. We also thank Ida Haugen Löfman and Michael Melin, cardiologists at the Huddinge transplant unit; Ronny Gustafsson, Shahab Nozohoor, Sandra Lindstedt, Johan Sjögren, and Richard Ingemansson, cardiac transplant surgeons at Skane University Hospital; Per Ola Kimblad and Lars Algotsson (former and present) head of the Department of Thoracic and Vascular Surgery at Skane University Hospital; and Ida Hedberg, Louise Del Treppo, and Jenny Warheim the research coordinators of the project. The authors also thank the cardiothoracic transplant coordinators, the personnel at the department of thoracic and vascular surgery and the Department of Heart and Lung Medicine at Skane University Hospital; and all members of our research group. The study was supported by the Swedish Research Council (2019-00487), Vinnova (2017-04689), Swedish Heart–Lung Foundation (20190623), a government grant for clinical research, region Skane research funds, donation funds from Skane University Hospital, the Anna-Lisa and Sven Eric Lundgrens Foundation, and Hans-Gabriel and Alice Trolle-Wachtmeister's Foundation for Medical Research. The supporting sources had no involvement in the study. Open access funding provided by Lund University.

## Author contributions

J.N.: principal investigator, study design, data interpretation, data analysis, writing of the paper, personnel education, and participation in the organ harvesting and transplantation. V.J.: study design, patient coordination, personnel education, data collection, and review of the final paper. G.Q.: participation in the heart explantation, attaching the donor hearts to the NIHP system, data collection, and review of the final paper. A.P.: participation in the donor organ explantation, responsible for the programming of the NIHP system, data collection, review of the final paper. C.M.: data interpretation, analysis and collection, and review of the final paper. T.S.: study design, study coordinator, review of the final paper. S.S.: study design, created the experimental models, and review of the final paper. All authors revised the paper for intellectual content and have seen and approved the final version.

## Competing interests

The device was patented by S.S. The patent is now owned by XVIVO Perfusion AB (Göteborg, Sweden). XVIVO Perfusion AB had no involvement in the study. The authors of this manuscript have no other competing interest to disclose.
