## [Peer Review File · Nature Communications]

Reviewers' comments:

Reviewer #1 (Remarks to the Author):

The manuscript continues to require reworking as a preliminary proof-of-concept that has no definitive clinical evidence, yet.

Reviewer #3 (Remarks to the Author):

Through revisions made to the manuscript from the previous submission to Nature Medicine, my comments have been addressed adequately.

Reviewer #4 (Remarks to the Author):

This study, a non randomised open label phase II study, is the first in man of the use of non-ischemic heart preservation (NIHP) for the preservation of hearts for transplantation. The current standard is static cold storage (SCS) and the time for heart preservation is limited to 6 hours, but NIHP can extend this to 24 hours. The outcomes of interest relate to safety.

This small study did not utilise the possible extended preservation time (actual mean times 223 minutes (range 202 – 263 minutes) for NIHP and 194 minutes (range 164-223 minutes) for the SCS).

This is not a randomised study and therefore although there has been comparison between the methods these cannot be taken to indicate efficacy of one group or the other. The results are all subject to bias. The statistical tests should be omitted. The results should be descriptive only.

The interest must centre on the safety of the new system. Therefore any statistical issues relate to the design rather than the statistical methods.

The design depending on looking at the characteristics of the two groups and the ensuing outcomes. The main issue, which requires a description, is how patients were selected for the NIHP group, which was limited to 6 patients, the others received usual care (SCS). This requires specialists in heart transplantation and the previous reviewers, whilst enthusiastic about the development of the new system, have provided an excellent review in explaining the clinical issues and moderating any claims made for the new system.

Abstract

There is no description of how the patients were allocated to the groups, the NIHP and the SCS. The issue appears to be how the 6 patients were chosen for the NIHP, and this should be described. Those not chosen got the routine SCS method. The abstract can report descriptive statistics but statistical testing is inappropriate. The abstract should also report that NIHP was not tested outside of the current usual time for heart preservation.

Response to the Reviewers comments:

Reviewer #1 (Remarks to the Author):

The manuscript continues to require reworking as a preliminary proof-of-concept that has no definitive clinical evidence, yet.

The manuscript has been refocused on safety and we have once again carried out a thorough review of all presented results and valued what these can have for clinical efficacy without overwhelming the conclusions. In the result section all p-values and “significant” have been deleted, only descriptive statistics are included. All conclusions and speculations indicating clinical evidence have been removed from the result section and toned down in the discussion. Finally, a description regarding patient selection for the NIHP system has been added to the first paragraph, page 5 (Recruitment).

Reviewer #3 (Remarks to the Author):

Through revisions made to the manuscript from the previous submission to Nature Medicine, my comments have been addressed adequately.

We thank the reviewer for all the previous comments which have helped us to improve the manuscript.

Reviewer #4 (Remarks to the Author):

This study, a non randomised open label phase II study, is the first in man of the use of non-ischemic heart preservation(NIHP) for the preservation of hearts for transplantation. The current standard is static cold storage (SCS) and the time for heart preservation is limited to 6 hours, but NIHP can extend this to 24 hours. The outcomes of interest relate to safety. This small study did not utilise the possible extended preservation time (actual mean times 223 minutes (range 202 – 263 minutes) for NIHP and 194 minutes (range 164-223 minutes) for the SCS.

We agree with the reviewer that would have been of interest to increase the preservation time, however the Ethical board did not accept that. The upper limit was 5.5 hours, which is also one criterion in the study protocol. The clinical practice in most centres today is to keep the ischemia below 4 hours. Already at 3 hours the mortality risk starts to increase. At 5 and 6 hours duration of ischemia the HR for one year mortality is 1.5 and 1.7, respectively according to the latest registry report (2019) from ISHLT (<https://ishltregistries.org/registries/slides.asp>). Finally, it would be unethical to evaluate new limits for preservation time in a safety study like this.

This is not a randomised study and therefore although there has been comparison between the methods these cannot be taken to indicate efficacy of one group or the other. The results are all subject to bias. The statistical tests should be omitted. The results should be descriptive only.

We thank the reviewer for this important comment and we have now deleted all p-values and updated the statistical paragraph. The outcomes in the second part of table 1 is now presented with descriptive statistics including 95% CI. To help the reader to value the difference between the groups the p-values have been exchanged to relative risk and effect size. The p-values in the figures have been omitted.

The interest must centre on the safety of the new system. Therefore any statistical issues relate to the design rather than the statistical methods. The design depending on looking at the characteristics of the two groups and the ensuing outcomes. The main issue, which requires a description, is how patients were selected for the NIHP group, which was limited to 6 patients, the others received usual care (SCS).

This requires specialists in heart transplantation and the previous reviewers, whilst enthusiastic about the development of the new system, have provided an excellent review in explaining the clinical issues and moderating any claims made for the new system.

We have now revised and added a detailed description regarding the donor and patient selection and the logistic behind the assignment of the NIHP device. The abstract has been updated by adding this two sentences: *“We used the same standard acceptance criteria for donor hearts for both groups. It was scheduled in advance when the NIHP method could be used, because the device and team members trained on the system must be available.”* Furthermore, the result section, paragraph 1, page 5, the description of the recruitment has been extended including how the NIHP system was selected. Finally, the Study logistic paragraph, page 12 in the Material and Method section has been extended including more details regarding the selection procedure.

“Assessment of potential donor heart was based on the usual constellation of clinical factors, including history, coronary angiography, echo assessment, and direct examination of the heart during procurement. The same standard criteria for donor hearts were used for the NIHP group. It was scheduled in advance when the NIHP method could be used, because the device and team members trained on the system need to be available. The donor and recipient characteristic, unless they met any exclusion criteria, did not exclude any patient to be assigned to the NIHP method. Furthermore, initially, we could only use ground transportation which limit the pool of potential donors. After the NIHP method was approved for air transport in April 2018, the system could be used with less restrictions, which resulting in that the NIHP method could be used in five of the total 11 transplantation performed over the next five months. Additionally, safety and logistic feasibility have to be assessed after the first and third patients where the NIHP method was used according to the local research ethics committee.”

Abstract

There is no description of how the patients were allocated to the groups, the NIHP and the SCS. The issue appears to be how the 6 patients were chosen for the NIHP, and this should be described. Those not chosen got the routine SCS method.

Two sentences of how the patients were allocated to the respectively groups have been added to the abstract, please also see previous answer.

The abstract can report descriptive statistics but statistical testing is inappropriate. The abstract should also report that NIHP was not tested outside of the current usual time for heart preservation.

The statistics have been omitted in the abstract. The median preservation time for both groups have been added.

REVIEWERS' COMMENTS:

Reviewer #3 (Remarks to the Author):

The manuscript is greatly improved. I have no further comments.

Reviewer #4 (Remarks to the Author):

The authors have addressed all my previous comments adequately. They have revised the manuscript providing extensive further details.

Response to the reviewer comments:

Reviewer #3 (Remarks to the Author):

The manuscript is greatly improved. I have no further comments.

We thank the reviewer for all the help to improve the manuscript.

Reviewer #4 (Remarks to the Author):

The authors have addressed all my previous comments adequately. They have revised the manuscript providing extensive further details.

We thank the reviewer for all the help to improve the manuscript.